# Multipole Approach to the Dynamical Casimir Effect with Finite-Size Scatterers

**DOI:** 10.3390/e26030251

**Published:** 2024-03-12

**Authors:** Lucas Alonso, Guilherme C. Matos, François Impens, Paulo A. Maia Neto, Reinaldo de Melo e Souza

**Affiliations:** 1Instituto de Física, Universidade Federal Fluminense, Niterói 24210-346, RJ, Brazil; lucasalonso@id.uff.br (L.A.); reinaldos@id.uff.br (R.d.M.e.S.); 2Instituto de Física, Universidade Federal do Rio de Janeiro, Rio de Janeiro 21941-972, RJ, Brazil; gcosta@pos.if.ufrj.br (G.C.M.); impens@if.ufrj.br (F.I.)

**Keywords:** dynamical Casimir effect, effective hamiltonians, multipolar expansion

## Abstract

A mirror subjected to a fast mechanical oscillation emits photons out of the quantum vacuum—a phenomenon known as the dynamical Casimir effect (DCE). The mirror is usually treated as an infinite metallic surface. Here, we show that, in realistic experimental conditions (mirror size and oscillation frequency), this assumption is inadequate and drastically overestimates the DCE radiation. Taking the opposite limit, we use instead the dipolar approximation to obtain a simpler and more realistic treatment of DCE for macroscopic bodies. Our approach is inspired by a microscopic theory of DCE, which is extended to the macroscopic realm by a suitable effective Hamiltonian description of moving anisotropic scatterers. We illustrate the benefits of our approach by considering the DCE from macroscopic bodies of different geometries.

## 1. Introduction

An oscillating neutral object in vacuum emits photons, the so-called dynamical Casimir effect (DCE)—see [1,2,3,4] and references therein. Photons are produced in pairs that satisfy the condition $\omega_{1}+\omega_{2}=\omega_{0},$ where $\omega_{1,2}$ denote the frequencies of the generated photons and $\omega_{0}$ is the mechanical frequency. The rate of photon production and its angular distribution were evaluated analytically for a perfectly conducting plate of infinite transverse size [5]. For a macroscopic mirror, $\omega_{0}/\left(2\pi \right)$ is typically below the MHz range, hence implying that the produced photons have wavelengths of the order or greater than $\lambda_{0}=2\pi c/\omega_{0}∼10^{2}\;m.$ Higher mechanical frequencies, in the GHz range, are achievable with thin-film resonators [6] and with plasmonic nanoantennas [7]. Recently, rotation frequencies beyond 5GHz were demonstrated with optically levitated nanoparticles [8]. In all cases, the transverse size of the movable surface is typically much smaller than $\lambda_{0}$. The dipole approximation then provides a much more realistic description than usual approaches based on the assumption of an infinite transverse size [5,9,10,11,12,13].

In this situation, the most convenient way to address the photon production is not by imposing boundary conditions on the moving plate [14] and deriving a relation between output and input fields [15,16,17] (for instance in terms of a Bogoliubov transformation [18]), but rather to employ directly a Hamiltonian approach [19,20,21,22]. Within the dipolar approximation, this strategy was successfully applied to evaluate the generation of photon pairs by an oscillating atom [23] in the microscopic dynamical Casimir effect (MDCE) [24,25,26,27,28,29,30,31]. In the present paper, we first revisit the MDCE effect by providing an alternative derivation of the associated Hamiltonian where the dipole motion gives rise to time-dependent higher-order multipole moments (Section 2). Our result allows us to consider anisotropic scatterers, which turns out to be a key element for extending the Hamiltonian approach into the macroscopic domain. General results for the DCE are then derived in Section 3, which are later applied to obtain the radiation emitted by a macroscopic metallic disk (Section 4). A comparison with the usual paradigm of an infinite oscillating surface reveals that finite-size effects can play a predominant role in the DCE. Indeed, the usual model of an infinite plate leads to an overestimation of the photon production rate by many orders of magnitude for any realistic configuration. Therefore, it is crucial to take into account the finite size of the oscillating mirror when estimating the required sensitivity in future DCE experiments with single objects in vacuum, which is precisely the main purpose of the present work. We also consider, in Section 5, an oscillating rod as a simpler illustration of the symmetries underlying the dynamical Casimir photons. Final remarks are presented in Section 6.

## 2. Effective Multipolar Hamiltonian for Moving Scatterers

In this Section, we shall consider a neutral polarizable object composed of several charged particles, with no permanent dipole, whose center-of-mass (CM) is set into a non-relativistic but otherwise arbitrarily prescribed motion r(t). The CM motion is always treated classically, and the electromagnetic response of the object may be strongly anisotropic. The considered object may be either microscopic (e.g., a molecule), or macroscopic (e.g., a disk mirror). Our approach rests on the electric dipole approximation, and thus applies as long as the relevant electromagnetic wavelengths are much larger than the system size—this will be the case for all examples considered throughout this paper. As mentioned in the introduction, this condition is also largely satisfied in realistic DCE experiments.

In an inertial frame where the dipole is instantaneously at rest, the Hamiltonian describing the interaction in the Schrödinger picture is $H^{\mathrm{rest}}=-d\cdot E^{\left(0\right)}\left(r\left(t\right)\right)$, where d is the dipole operator of the system and $E^{\left(0\right)}\left(r\left(t\right)\right)$ is the electric field operator evaluated at the dipole CM position. We may obtain the corresponding laboratory-frame Hamiltonian by using a Lorentz boost to relate the field to different frames, as is usually carried out in the literature. However, here we shall follow an alternative path by recognizing that a moving dipole can always be described in terms of multipoles which are fixed at the origin of the coordinate system. In order to do so, we express the Hamiltonian by considering the expansion up to the electric quadrupole and magnetic dipole order. We start with a classical description and subsequently follow a standard procedure to quantize it.

We start with the usual minimal coupling framework, where the Lagrangian density for the matter–light interaction is given by
(1)$$L_{\mathrm{int}}\left(r,t\right)=-\frac{1}{c}j_{\mu }A^{\mu }=-\rho \left(r,t\right)\phi \left(r,t\right)+\frac{1}{c}j\left(r,t\right)\cdot A\left(r,t\right)\;,$$
where $j^{\mu }=\left(\rho c,j\right)$ is the four-charge current and $A^{\mu }=\left(\phi ,A\right)$ is the four potential in Gaussian units. Let us assume that our charge distribution is neutral and may be treated within the multipolar approximation up to the electric quadrupole and dipole magnetic terms. We may derive the multipolar Lagrangian from Equation (Equation 1) following standard procedures [32] of adding a total time derivative, but it will be simpler to follow an alternative route by explicitly employing the charge and current density associated with the multipoles [33]:(2)$$\begin{array}{ccc} \rho (r,t) & = & -d_{j}\left(t\right)\partial_{j}\delta \left(r\right)+Q_{jk}\left(t\right)\partial_{j}\partial_{k}\delta \left(r\right) \end{array}$$(3)$$\begin{array}{ccc} j_{k}\left(r,t\right) & = & {\dot{d}}_{k}\left(t\right)\delta \left(r\right)-T_{kl}\left(t\right)\partial_{l}\delta \left(r\right), \end{array}$$
where, from now on, we use Einstein’s notation by summing over repeated indices. $d_{i}\left(t\right)=\int \rho \left(r,t\right)r_{i}d^{3}r$ and $Q_{ij}\left(t\right)=\frac{1}{2}\int \rho \left(r,t\right)r_{i}r_{j}d^{3}r$ represent the components of the electric dipole and quadrupole moments, respectively, while $T_{kl}\left(t\right)=\int j_{k}\left(r,t\right)r_{l}d^{3}r$ contains the second-order correction to the electric current. Its symmetric part $T_{kl}\left(t\right)+T_{lk}\left(t\right)=2{\dot{Q}}_{kl}\left(t\right)$ captures the electric quadrupole’s contribution to the current, while its antisymmetric part can be written as the dual tensor to the magnetic dipole moment $m\left(t\right)=\frac{1}{2}\int r\times j\left(r,t\right)d^{3}r$, namely $T_{kl}\left(t\right)-T_{lk}\left(t\right)=2\epsilon_{kln}m_{n}\left(t\right)$ ($\epsilon_{kln}$ is the Levi-Civita symbol).

Let us evaluate the multipole moments of our moving system. We seek a general description which is valid for any motion compatible with the dipole approximation, i.e., such that the system emits mostly in wavelengths much larger than its size. We thus take as our exact charge distribution $\rho \left(r,t\right)=-d_{0}\left(t\right)\cdot \nabla \delta \left(r-r_{0}\left(t\right)\right)$, where $r_{0}\left(t\right)$ represents the position of an arbitrary point O attached to the system and $d_{0}\left(t\right)$ is the electric dipole moment of our moving system with respect to O. As a consistency check, this charge distribution yields an electric dipole moment $d\left(t\right)=d_{0}\left(t\right)$, independently of the system position as expected for a neutral system. Consequently, higher-order multipole moments carry all the signatures of dynamical motion. For the same reason we do not have to worry about the intrinsic electric quadrupole and magnetic dipole moments of our system as their dynamical corrections will manifest only in the electric octopole and magnetic quadrupole terms, and thus can be neglected.

In order to evaluate the tensor $T_{ki}\left(t\right)$, we first write the electric current. For the considered model, it is given by $j\left(r,t\right)=-v\left(t\right)d_{0}\left(t\right)\cdot \nabla \delta \left(r-r_{0}\left(t\right)\right)+{\dot{d}}_{0}\left(t\right)\delta \left(r-r_{0}\left(t\right)\right)$, where $v={\dot{r}}_{0}$ is the moving dipole velocity. The first term is nothing but the usual ρv contribution to the current. The second term is the polarization current, originating from the time variation in the electric dipole, which ensures consistency with the local charge conservation $\partial_{t}\rho +\nabla \cdot j=0$. One obtains
(4)$$T_{kl}={\dot{d}}_{0k}r_{0l}+v_{k}d_{0l}\;.$$

We work in the Coulomb gauge and set ϕ=0, which amounts to discard the self-energies of the multipoles [34]. We note from now on that $d_{0}\left(t\right)\equiv d\left(t\right)$ and integrate Equation (Equation 1) to write the interaction Lagrangian as
(5)$$L_{\mathrm{int}}\left(t\right)=\int L_{\mathrm{int}}\left(r,t\right)d^{3}r={\dot{d}}_{k}\left(t\right)\frac{A_{k}\left(0,t\right)}{c}+{\dot{d}}_{k}\left(t\right)r_{0l}\left(t\right)\frac{\partial_{l}A_{k}\left(0,t\right)}{c}+\frac{v_{k}}{c}\left(t\right)d_{l}\left(t\right)\partial_{l}A_{k}\left(0,t\right)\;,$$
by virtue of Equations (Equation 3) and (Equation 4). Adding a total time derivative to the Lagrangian does not affect the equations of motion for the field. Thus, the transformation $L_{\mathrm{int}}\left(t\right)\to L_{\mathrm{int}}\left(t\right)-\frac{1}{c}\frac{dh(t)}{dt}$ with $h\left(t\right)=d_{k}\left(t\right)A_{k}\left(t\right)+{\dot{d}}_{k}\left(t\right)r_{0l}\left(t\right)\partial_{l}A_{k}\left(t\right)$ generates the equivalent Lagrangian
(6)$$L_{\mathrm{int}}\left(t\right)=d_{k}\left(t\right)E_{k}\left(0,t\right)+d_{k}\left(t\right)r_{0l}\left(t\right)\partial_{l}E_{k}\left(0,t\right)-d_{k}\left(t\right)\frac{v_{l}\left(t\right)}{c}\partial_{l}A_{k}\left(0,t\right)+\frac{v_{k}\left(t\right)}{c}d_{l}\left(t\right)\partial_{l}A_{k}\left(0,t\right)\;,$$
where $E=-\dot{A}/c$ is the electric field. By combining the last two terms (with an exchange of the summation indices), one obtains a magnetic contribution of the form $v_{l}d_{k}\left(\partial_{k}A_{l}-\partial_{l}A_{k}\right)=\epsilon_{klm}v_{l}d_{k}B_{m}$. One can then express the Lagrangian in terms of the EM fields as
(7)$$L_{\mathrm{int}}\left(t\right)=d\left(t\right)\cdot \left[E\left(0,t\right)+r_{0}\left(t\right)\cdot \nabla E\left(r,t\right){|}_{r=0}+\frac{v(t)}{c}\times B\left(0,t\right)\right]\;.$$
We may now follow the usual quantization procedure [34] and write the interaction Hamiltonian in the Schrödinger picture
(8)$$H_{\mathrm{int}}\left(t\right)=-d\cdot \left[E\left(0\right)+r_{0}\left(t\right)\cdot \nabla E\left(r\right){|}_{r=0}+\frac{v(t)}{c}\times B\left(0\right)\right]\;.$$
From now on d is an operator, as well as the electric and magnetic fields, given explicitly by
(9)$$\begin{array}{ccc} E(r) & = & \sum_{k\sigma }i\sqrt{\frac{2\pi \hbar \omega_{k}}{V}}{\hat{\epsilon }}_{k\sigma }\left(a_{k\sigma }e^{ik\cdot r}-a_{k\sigma }^{†}e^{-ik\cdot r}\right) \end{array}$$
(10)$$\begin{array}{ccc} B(r) & = & \sum_{k\sigma }ic\sqrt{\frac{2\pi \hbar }{\omega_{k}V}}k\times {\hat{\epsilon }}_{k\sigma }\left(a_{k\sigma }e^{ik\cdot r}-a_{k\sigma }^{†}e^{-ik\cdot r}\right)\;, \end{array}$$
where *V* is the quantization volume (we shall take V→+∞ in the end), $\omega_{k}=c\left|k\right|$, $a_{k\sigma }$($a_{k\sigma }^{†}$) is the annihilation (creation) operator for the mode (k,σ), and ${\hat{\epsilon }}_{k\sigma }$ is the unit vector for polarization σ, assumed to be real without the loss of generality.

The Hamiltonian (Equation 8) acts on a quantum system composed of the internal degrees of freedom (dofs) of the scatterer and of the electromagnetic field’s dofs. Its explicit time dependence accounts for the coupling of this quantum system with the center-of-mass motion (treated here classically), which drives the DCE radiation process and provides the corresponding energy. We consider from now on a prescribed harmonic motion of the form $r_{0}\left(t\right)=a\cos\left(\omega_{\mathrm{cm}}t\right).$ The relevant (i.e., significantly populated) DCE mode frequencies lie in the interval $\left[0,\omega_{\mathrm{cm}}\right]$.

Let us discuss the magnitude of the successive terms of the interaction Hamiltonian (Equation 8), noting for convenience $H_{\mathrm{int}}^{E0}=-d\cdot E\left(0\right)$, $H_{\mathrm{int}}^{E1}=-d\cdot \left(r_{0}\left(t\right)\cdot \nabla E\left(r\right){|}_{r=0}\right)$ and $H_{\mathrm{int}}^{B1}=-d\cdot \left(\frac{v(t)}{c}\times B\left(0\right)\right)$. One finds that $\left|\right|H_{\mathrm{int}}^{E1}\left|\right|/\left|\right|H_{\mathrm{int}}^{E0}\left|\right|∼\omega_{k}a/c\leq \omega_{\mathrm{cm}}a/c$. Thus, the term $H_{\mathrm{int}}^{E1}$ appears as a first-order relativistic correction in the scatterer velocity $v∼\omega_{\mathrm{cm}}a$ to the static dipole interaction. A similar argument holds for the Röntgen term $H_{\mathrm{int}}^{B1}$: $\left|\right|H_{\mathrm{int}}^{B1}\left|\right|/\left|\right|H_{\mathrm{int}}^{E0}\left|\right|∼v/c$. Similarly, one finds that the successive multipole contributions (capturing the dynamical motion of the dipole) are scaled as increasing powers of (v/c). Working up to the first nonrelativistic order, one retrieves the usual interaction Hamiltonian $H_{\mathrm{int}}^{\left(1\right)}=-d\cdot \left[E+v\times B/c\right]$ corresponding to a Lorentz transformation of the electric field to the instantaneous rest frame.

Let us now comment on the specific internal quantum structure of the moving scatterer. For atoms and molecules, the frequency of the prescribed mechanical harmonic motion is typically too small to excite internal transitions. On the other hand, low-frequency internal excitation channels are usually available in macroscopic scatterers made of a dissipative material, as for instance, a metallic mirror. Such internal dissipation channels allow for a quantum friction between scatterers in relative motion in a vacuum [35], which we disregard since our focus is the dynamical Casimir effect for a single moving scatterer. Thus, we assume that the internal dofs remain in their ground state throughout the motion so that photons can only be produced in pairs due to the second order of perturbation theory in the Hamiltonian (Equation 8).

Alternatively, we can build up an effective Hamiltonian which takes into account, by construction, the virtual internal transitions of the system. Using Equations (Equation 9) and (Equation 10), we write the interaction Hamiltonian (Equation 8) as $H_{\mathrm{int}}\left(t\right)=-d\left(t\right)\cdot F\left(t\right),$ with the field operator F(t) given in terms of time-dependent coefficients $F_{k\sigma }\left(t\right)$ as follows:(11)$$F\left(t\right)=\sum_{k\sigma }F_{k\sigma }\left(t\right)\left(a_{k\sigma }-a_{k\sigma }^{†}\right)\;.$$

This allows us to perform the same unitary transform as in Ref. [36] and work with the effective Hamiltonian
(12)$$H_{\mathrm{eff}}=-\frac{1}{2}\sum_{k\sigma }\alpha_{ij}\left(\omega_{k}\right)F_{i,k\sigma }\left(t\right)\left(a_{k\sigma }-a_{k\sigma }^{†}\right)F_{j}\left(t\right)\;,$$
where $\alpha_{ij}\left(\omega \right)$ is the polarizability of the scatterer:(13)$$\alpha_{ij}\left(\omega \right)=\frac{2}{\hbar }\sum_{e}\frac{\omega_{eg}\langle g|d_{i}\left|e\rangle \langle e\right|d_{j}|g\rangle }{\left(\omega_{eg}^{2}-\omega^{2}\right)}\;.$$
Here, |g〉 and |e〉 denote the (internal) ground and excited states, respectively, and $\omega_{eg}=\left(E_{e}-E_{g}\right)/\hbar$ stands for the transition frequency.

The effective Hamiltonian (Equation 12) generalizes the approach of Refs. [36,37,38] to the case of moving scatterers. It has the convenience of capturing two-photon processes within a first-order perturbation theory description of the coupling between a moving neutral quantum system, polarizable but with no permanent dipole, and the quantum electromagnetic field. As mechanical frequencies are typically much smaller than the internal transition frequencies, we are entitled to neglect dispersion and take the static polarizability: $\alpha_{ij}\left(\omega_{k}\right)\approx \alpha_{ij}\left(0\right)$. With this approximation and employing Equation (Equation 11) we obtain
(14)$$H_{\mathrm{eff}}\approx -\frac{1}{2}\alpha_{ij}\left(0\right)\left[E_{i}\left(0\right)E_{j}\left(0\right)+\left(r_{0}\left(t\right)\cdot \nabla \right)\left(E_{i}\left(r\right)E_{j}\left(r\right)\right){|}_{r=0}-2\frac{v_{l}\left(t\right)}{c}\epsilon_{lnj}E_{i}\left(0\right)B_{n}\left(0\right)\right]\;,$$
where we have neglected higher-order contributions involving second spatial derivatives of order ${\left(v/c\right)}^{2}.$ In the last term we have used the symmetry of the polarizability tensor as well as the fact that fields components at the same point commute with each other, as can be appreciated in Equations (Equation 9) and (Equation 10). The effective Hamiltonian (Equation 14) is not restricted to microscopic quantum systems. It also captures the radiation of moving macroscopic scatterers treated in the dipole approximation, and will be our departure point for investigating illustrative examples of DCE in the next sections.

## 3. Photon Emission Rate

In this section we derive the dynamical Casimir photon emission rate from the effective Hamiltonian (Equation 14). As already mentioned in the previous section, we consider that our scatterer is moving harmonically according to $r_{0}\left(t\right)=a\cos\left(\omega_{\mathrm{cm}}t\right)$, where a is a constant vector denoting the amplitude of the motion. We choose to express the amplitude in the form $a=-\frac{v}{\omega_{\mathrm{cm}}}\hat{u}$, where the constant *v* denotes the maximum velocity achieved by the scatterer. The Hamiltonian (Equation 14) then assumes the form
(15)$$H_{\mathrm{eff}}=-\frac{1}{2}\alpha_{ij}\left(0\right)E_{i}\left(0\right)E_{j}\left(0\right)+V_{1}\cos\left(\omega_{\mathrm{cm}}t\right)+V_{2}\sin\left(\omega_{\mathrm{cm}}t\right)\;.$$

The operators $V_{1}$ and $V_{2}$ capture the spatial variation in the electric field and the Röntgen current, respectively. As derived in the previous section, they arise as combinations of magnetic dipole and electric quadrupole contributions resulting from the motion of the scatterer considered within the electric dipole approximation and are given by
(16)$$\begin{array}{ccc} V_{1} & = & \frac{v}{2\omega_{0}}\alpha_{ij}\left(0\right)\hat{u}\cdot \nabla \left(E_{i}\left(r\right)E_{j}\left(r\right)\right){|}_{r=0}\;, \end{array}$$
(17)$$\begin{array}{ccc} V_{2} & = & -\alpha_{ij}\left(0\right)\frac{v}{c}\epsilon_{lnj}{\hat{u}}_{l}E_{i}\left(0\right)B_{n}\left(0\right)\;. \end{array}$$
Finally, note that, in the particular case of isotropic scatters, $\alpha_{ij}\left(0\right)=\alpha_{0}\delta_{ij},$ (Equation 15)–(17) agree with Ref. [23].

A great convenience of our effective Hamiltonian is that it involves only field dofs. Within first-order perturbation theory, only two-photon states can be generated from the initial vacuum state; a well-known feature of DCE: photons are generated in pairs. More concretely, the field state reads as
(18)$$|\psi \left(t\right)\rangle =|0\rangle +\sum_{\sigma ,\sigma^{\prime }}\int \frac{d^{3}k}{{\left(2\pi \right)}^{3}}\int \frac{d^{3}k^{\prime }}{{\left(2\pi \right)}^{3}}\;c_{k\sigma k^{\prime }\sigma^{\prime }}\left(t\right)|1_{k\sigma }1_{k^{\prime }\sigma^{\prime }}\rangle \;,$$
where |0〉 denotes the vacuum state of the electromagnetic field, while $|1_{k\sigma }1_{k^{\prime }\sigma^{\prime }}\rangle$ denotes a state with two photons; one in mode (k,σ) and the other in mode $\left(k^{\prime },\sigma^{\prime }\right)$. We are interested in the long time limit $t\gg 1/\omega_{\mathrm{cm}}$, where Fermi’s golden rule implies that $H_{\mathrm{eff}}$ connects only states differing by $\hbar \omega_{\mathrm{cm}}$ in energy. Denoting the frequencies of the produced photons by ω=|k|c and $\omega^{\prime }={\left|k\right|}^{\prime }c$, we have
(19)$$\omega +\omega^{\prime }=\omega_{\mathrm{cm}}\;.$$
This is a remarkable feature of a sinusoidal perturbation, since the center of mass is here treated as classical and nonetheless exchanges energy only in the *quanta* of $\hbar \omega_{\mathrm{cm}}$, which is analogous to what happens in the photoelectric effect, which can be explained by a classical description of the electromagnetic field [39]. We find the photon emission rate for a given pair of photons $\left(k\sigma ,k^{\prime }\sigma^{\prime }\right)$ from the corresponding amplitude for pair production:(20)$$\frac{d\Gamma_{\sigma ,\sigma^{\prime }}\left(k,k^{\prime }\right)}{d^{3}kd^{3}k^{\prime }}=\frac{|c_{k\sigma k^{\prime }\sigma^{\prime }}{|}^{2}}{t}.$$
Providing that the polarizability tensor is symmetric, we find
(21)$$\frac{d\Gamma_{\sigma ,\sigma^{\prime }}\left(k,k^{\prime }\right)}{d^{3}kd^{3}k^{\prime }}=\frac{\omega \omega^{\prime }v^{2}}{32\pi^{3}c^{2}}{\left\{\alpha_{ij}\left[-\hat{u}\cdot \left(x^{\prime }\hat{k}+x{\hat{k}}^{\prime }\right){\hat{\epsilon }}_{k\sigma }^{i}{\hat{\epsilon }}_{k^{\prime }\sigma^{\prime }}^{j}+{\hat{u}}_{l}\left({\hat{k}}_{j}{\hat{\epsilon }}_{k\sigma }^{l}{\hat{\epsilon }}_{k^{\prime }\sigma^{\prime }}^{i}+{\hat{k}}_{j}^{\prime }{\hat{\epsilon }}_{k^{\prime }\sigma^{\prime }}^{l}{\hat{\epsilon }}_{k\sigma }^{i}\right)\right]\right\}}^{2}\delta \left(\Delta \omega \right)\;,$$
where $x=\omega /\omega_{\mathrm{cm}}=1-x^{\prime }$ and $\Delta \omega =\omega +\omega^{\prime }-\omega_{\mathrm{cm}}.$ The first term within brackets in (Equation 21), arising from the operator $V_{1},$ contains a non-trivial frequency dependence associated with *x* and $x^{\prime }$ that results from taking the gradient of the electric field operators in (Equation 16). The second term, proportional to ${\hat{u}}_{l},$ captures the contribution of the Röntgen operator $V_{2}$ and does not introduce any additional frequency dependence apart from the prefactor $\omega \omega^{\prime }$ which comes from the density of states. In a case where the scatterer is randomly rotating in a time scale much smaller than the typical emission time, we may substitute the polarizability tensor with its average $\alpha_{ij}=\bar{\alpha }\delta_{ij}$, with $\bar{\alpha }=\alpha_{ii}/3$. In this case, we reobtain the result for an isotropic system [23].

We may integrate Equation (Equation 21) with $d^{3}k^{\prime }$ and sum with $\sigma^{\prime }$ in order to obtain the angular spectral distribution of the DCE radiation. This can be readily carried out by employing the symmetrical properties of the rotation group, as detailed in appendix B of ref. [23], which yields the result
(22)$$\frac{d\Gamma_{\sigma }\left(k\right)}{d\omega d\Omega }=\frac{\omega^{3}{\left(\omega_{\mathrm{cm}}-\omega \right)}^{3}v^{2}}{60\pi^{2}c^{8}}\alpha_{ij}\alpha_{mn}P_{ijmn}\;,$$
where we have written $d^{3}k=\frac{\omega^{2}}{c^{3}}d\omega d\Omega$ and
(23)$$P_{ijmn}=5{\hat{u}}_{r}{\hat{u}}_{l}\left(x^{\prime 2}{\hat{k}}_{r}{\hat{k}}_{l}{\hat{\epsilon }}_{k\sigma }^{j}{\hat{\epsilon }}_{k\sigma }^{n}-2x^{\prime }{\hat{k}}_{r}{\hat{k}}_{n}{\hat{\epsilon }}_{k\sigma }^{j}{\hat{\epsilon }}_{k\sigma }^{l}+{\hat{k}}_{j}{\hat{k}}_{n}{\hat{\epsilon }}_{k\sigma }^{r}{\hat{\epsilon }}_{k\sigma }^{l}\right)\delta_{im}+{\hat{\epsilon }}_{k\sigma }^{j}{\hat{\epsilon }}_{k\sigma }^{n}\left[\delta_{im}\left(2x^{2}+x+2\right)-{\hat{u}}_{i}{\hat{u}}_{m}\left(x^{2}+3x+1\right)\right]\;.$$
We emphasize that expression (Equation 22) is valid only for $\omega \leq \omega_{\mathrm{cm}}$. Indeed, the photon production rate vanishes outside this interval to become first order in the perturbation as it is not possible to satisfy condition (Equation 19). Summation over the polarization and integration over the solid angle yields the frequency spectral rate for photon production:(24)$$\frac{d\Gamma }{d\omega }=\frac{2\omega_{\mathrm{cm}}^{6}v^{2}}{45\pi c^{8}}{\left[\left(1-x\right)x\right]}^{3}\left\{\left[7-4x\left(1-x\right)\right]\alpha_{ij}\alpha_{ij}-2\left[3-x\left(1-x\right)\right]\alpha_{ij}\alpha_{in}{\hat{u}}_{j}{\hat{u}}_{n}\right\}\;.$$
As expected, since the photons are generated in pairs satisfying condition (Equation 19), the spectrum is symmetrical around $\omega =\omega_{\mathrm{cm}}/2.$ Indeed, the spectrum (Equation 24) is invariant under the exchange x↔(1−x). Also, it vanishes at ω=0, as it should since the electromagnetic field density of states vanishes in this case. Given the symmetry of the spectrum, it also vanishes at $\omega =\omega_{\mathrm{cm}}.$ Naturally, the right-hand-side of Equation (Equation 24) is positive definite in the physical region 0≤x≤1, as can be immediately verified by noticing that $\alpha_{ij}\alpha_{in}{\hat{u}}_{j}{\hat{u}}_{n}\leq \alpha_{ij}\alpha_{ij}$ (this inequality is readily established once the (symmetrical) polarizability tensor is expressed in its principal-axes basis). Photon production is maximized when the vector $\alpha_{ij}{\hat{u}}_{j}$ has the smallest magnitude, which happens when the motion is aligned along the principal axis of the polarizability tensor with the smallest eigenvalue.

Finally, the total photon production rate is given by
(25)$$\Gamma =\frac{\omega_{\mathrm{cm}}^{7}v^{2}}{5670\pi c^{8}}\left(11\alpha_{ij}\alpha_{ij}-10\alpha_{ij}\alpha_{in}{\hat{u}}_{j}{\hat{u}}_{n}\right)\;.$$
Let us consider an almost isotropic scatterer, where the principal values of the polarizability are given by $\alpha^{\left(i\right)}=\bar{\alpha }+\delta \alpha^{\left(i\right)}$, with $|\delta \alpha^{\left(i\right)}|\ll \bar{\alpha }$ and $\sum_{i}\delta \alpha^{\left(i\right)}=0$. We consider the motion to be parallel to the principal axis *j* of the scatterer, so that to first-order expression (Equation 25) simplifies to
(26)$$\Gamma =\frac{\omega_{\mathrm{cm}}^{7}v^{2}}{5670\pi c^{8}}\left(23\bar{\alpha }-20\bar{\alpha }\delta \alpha^{\left(j\right)}\right)\;.$$
We see that photon production is enhanced by a small anisotropy which diminishes the polarization along the direction of motion ($\delta \alpha^{\left(j\right)}<0$). In the following sections we shall examine in more detail two cases in which the anisotropy is not small.

## 4. An Oscillating Disk

Let us consider now a neutral macroscopic metallic mirror oscillating with frequency $\omega_{\mathrm{cm}}$. We shall assume the mirror to be a very thin disk of radius *R*. We consider here the typical situation where $\omega_{\mathrm{cm}}R/c\ll 1$, enabling us to treat it within the dipole approximation. We also assume that $\omega_{\mathrm{cm}}$ is much smaller than the typical frequency scales associated with the metallic medium of which the mirror is made, such as the plasma frequency. We may then employ the expressions derived in the last section, which are written in terms of the static polarizability tensor. We take the *z*-axis along the symmetry axis of the mirror. With this choice, the polarizability tensor is given by [40]
(27)$$\overset{↔}{\alpha }=\frac{4R^{3}}{3\pi }\left(\begin{array}{ccc} 1 & 0 & 0\\ 0 & 1 & 0\\ 0 & 0 & 0 \end{array}\right)\;.$$
Notice that the dipole induced by fields parallel to the mirror scales with $R^{3}$, rather than with the volume of the mirror, which contains a much smaller length scale associated with its width.

### 4.1. Motion Perpendicular to the Plate

We begin with the case where $\hat{u}=\hat{z}$, as depicted in Figure 1a.

As discussed in Section 3, this will be the case with the greatest photon production. In addition, this is usually the situation of practical interest. Nonetheless, we emphasize that photons are generated even if the mirror oscillates along a direction perpendicular to its symmetry axis, which is not the case when considering a perfectly conducting mirror to be infinite.

The angular spectral distribution for the photon production is given by Equation (Equation 22) with the polarizability tensor given by Equation (Equation 27). We take the usual Fresnel-like polarization basis in order to analyze each polarization separately. We define ${\hat{\epsilon }}_{k}^{\left(\mathrm{TE}\right)}=\hat{z}\times \hat{k}/\left|\hat{z}\times \hat{k}\right|$ as the unitary vector for transverse electric (TE) polarization and ${\hat{\epsilon }}_{k}^{\left(\mathrm{TM}\right)}={\hat{\epsilon }}_{k}^{\left(\mathrm{TE}\right)}\times \hat{k}$ for the transverse magnetic (TM) one. With this choice, Equation (Equation 22) becomes
(28)$$\frac{d\Gamma_{\sigma }^{⊥}\left(k\right)}{d\omega d\Omega }=\frac{4\omega^{3}{\left(\omega_{\mathrm{cm}}-\omega \right)}^{3}v^{2}R^{6}}{135\pi^{4}c^{8}}f_{\sigma }^{⊥}\left(\frac{\omega }{\omega_{\mathrm{cm}}},\theta \right)\;,$$
where θ is the angle between k and $\hat{z}$ and
(29)$$\begin{array}{ccc} f_{\mathrm{TE}}^{⊥}\left(x,\theta \right) & = & 5\cos^{2}\theta {\left(1-x\right)}^{2}+\left(2x^{2}+x+2\right) \end{array}$$
(30)$$\begin{array}{ccc} f_{\mathrm{TM}}^{⊥}\left(x,\theta \right) & = & 5{\left(1-x\cos^{2}\theta \right)}^{2}+\cos^{2}\theta \left(2x^{2}+x+2\right)\;. \end{array}$$
Note that, for θ=0,π, we have $f_{\mathrm{TE}}^{⊥}=f_{\mathrm{TM}}^{⊥}$, as we should, since in this case of axial symmetry the two polarizations are equivalent. Notice also that the angular spectrum for TE-polarized photons favors emission along the direction of motion but becomes more isotropic as the photon frequency approaches $\omega_{\mathrm{cm}}$ (x→1). The spectrum for TM polarization has a more complex dependence on the angle of emission. Small-frequency TM photons are emitted mostly along the direction of motion. On the other hand, directions perpendicular to the motion are favored by $\omega >2\omega_{\mathrm{cm}}/7,$ as can be shown by solving the second-degree inequality for *x* arising from the condition $f_{\mathrm{TM}}^{⊥}\left(x,\pi /2\right)>f_{\mathrm{TM}}^{⊥}\left(x,0\right)$. In Figure 2a,b we present the polar plots for $\omega =\omega_{\mathrm{cm}}/2$.

By performing the angular integration of (Equation 28), we find the frequency spectrum for each polarization:(31)$$\begin{array}{c} \frac{d\Gamma_{\sigma }^{⊥}}{d\omega }=\frac{16\omega^{3}{\left(\omega_{\mathrm{cm}}-\omega \right)}^{3}v^{2}R^{6}}{405\pi^{3}c^{8}}F_{\sigma }^{⊥}\left(\frac{\omega }{\omega_{\mathrm{cm}}}\right)\;, \end{array}$$
with
(32)$$\begin{array}{ccc} F_{\mathrm{TE}}^{⊥}\left(x\right) & = & 11x^{2}-7x+11 \end{array}$$
(33)$$\begin{array}{ccc} F_{\mathrm{TM}}^{⊥}\left(x\right) & = & 5x^{2}-9x+17\;. \end{array}$$

The spectra are plotted in Figure 3. We have more TM photons than TE ones for any value of frequency. TE photons are preferably emitted with a frequency greater than $\omega_{\mathrm{cm}}$, with the opposite happening for the TM polarization. When we add the two polarizations, we have a spectrum symmetric around $\omega =\omega_{\mathrm{cm}}/2$, in agreement with the general property discussed in Section 3. The total photon production rates are given by
(34)$$\Gamma_{\mathrm{TM}}^{⊥}=\frac{20\omega_{\mathrm{cm}}^{7}v^{2}R^{6}}{5103\pi^{3}c^{8}}=\frac{25}{19}\Gamma_{\mathrm{TE}}^{⊥}\;.$$

Let us compare our results with the ones obtained in Ref. [5] by treating the plate as having an infinite extension. As might be expected, the infinite plate assumption greatly overestimates the photon production rate. While the photon production rate for a finite-size mirror is proportional to the square of its polarizability, and thus to $R^{6}$, the infinite plate approximation leads to a photon production per unit area which is independent of geometrical factors, and thus the photon production rate scales with $R^{2}$ in this case. Indeed, the infinite plate calculation overestimates the photon production rate by $382,725\pi^{2}{\left(c/\left(\omega_{\mathrm{cm}}R\right)\right)}^{4}$ which is in the order of ∼$10^{24}$ for typical values *R*∼ 1 cm and $\omega_{\mathrm{cm}}$ ∼ 1 MHz.

Another striking difference between the two cases is the lack of translational symmetry parallel to the surface of a finite-size mirror. On the other hand, such symmetry holds for an infinite plate and enforces correlations between the photons in a given pair. They must (i) have the same polarization and (ii) opposite projections of their wavevectors parallel to the plate. In our case we have none of these restrictions, as can be appreciated in Equation (Equation 21). The absence of restriction (i) implies that the spectrum for each separate polarization is not symmetrical around $\omega =\omega_{\mathrm{cm}}/2$, as discussed in connection with Figure 3. The absence of restriction (ii) is the major reason for the qualitative differences between the angular distributions for the finite-size and infinite-size mirrors. Finally, for an infinite plate both the angular and frequency spectra for TM photons present a logarithm divergence due to the excitation of resonant surface plasmons, which is naturally absent from our results since there is no analogue version of a long-range coherent oscillation when considering a single dipole scatterer. Aside from their many differences, two similarities are worth noticing. TE photons are mostly emitted along the direction of motion for any frequency. In addition, more TM than TE photons are produced also for any frequency, although the difference is much smaller in the case of a finite size mirror.

### 4.2. Motion Parallel to the Plate

In this case we do not have axial symmetry. In addition to θ, we shall define ϕ as the angle between the projection of k in the plane of the disk and the unit vector $\hat{u}$ pointing along the direction of motion, now assumed to be in the plane of the disk along the *x* direction, as illustrated in Figure 1b.

The angular spetrum is now given by
(35)$$\frac{d\Gamma_{\sigma }^{\|}\left(k\right)}{d\omega d\Omega }=\frac{4\omega^{3}{\left(\omega_{\mathrm{cm}}-\omega \right)}^{3}v^{2}R^{6}}{135\pi^{4}c^{8}}f_{\sigma }^{\|}\left(\frac{\omega }{\omega_{\mathrm{cm}}},\theta ,\phi \right)\;,$$
with
(36)$$\begin{array}{ccc} f_{\mathrm{TE}}^{\|}\left(x,\theta ,\phi \right) & = & 5\sin^{2}\theta \cos^{2}\phi {\left(1-x\right)}^{2}+5\sin^{2}\theta \sin^{2}\phi +{\left(1-x\right)}^{2}+\cos^{2}\phi \left(x^{2}+3x+1\right) \end{array}$$
(37)$$\begin{array}{ccc} f_{\mathrm{TM}}^{\|}\left(x,\theta ,\phi \right) & = & 5\sin^{2}\theta \cos^{2}\theta \cos^{2}\phi x^{2}+\cos^{2}\theta {\left(1-x\right)}^{2}+\cos^{2}\theta \sin^{2}\phi \left(x^{2}+3x+1\right)\;. \end{array}$$
At θ=0, the factor $\cos^{2}\phi$ ($\sin^{2}\phi$) which remains in $f_{\mathrm{TE}}^{\|}$ ($f_{\mathrm{TM}}^{\|}$) arises entirely from the expressions for the TE and TM polarization unit vectors in the limit θ→0. Indeed, the unit vector perpendicular to the direction of motion on the xy plane reads $\hat{y}=\cos\phi \;{\hat{\epsilon }}_{k}^{\left(\mathrm{TE}\right)}+\sin\phi \;{\hat{\epsilon }}_{k}^{\left(\mathrm{TM}\right)}$ in this limit. The dependence on ϕ then simply indicates that the dynamical Casimir radiation emitted along the direction normal to the plate is partially polarized perpendicularly to the motion of the plate.

In Figure 2c we present polar plots for TE polarization and $\omega =\omega_{\mathrm{cm}}/2$. TE-polarized photons are mostly emitted along directions parallel to the mirror, and especially perpendicularly to the direction of motion for $\omega >\omega_{\mathrm{cm}}/6$. For the opposite case of low frequency photons, emission is greater along the direction of motion.

As illustrated by Figure 2d, TM-polarized photons are not emitted along a direction parallel to the plane of the disk (θ=π/2). This is due to the fact that the xy plane containing the disk is a symmetry plane with respect to space reflection. Symmetry then requires that the vector (pseudovector) fields, like the electric (magnetic) one, must lie parallel (perpendicular) to the xy plane. Thus, only TE-polarized photons can be emitted along directions corresponding to θ=π/2. There are more TM photons emitted in the plane ϕ=π/2 (perpendicular to the motion) than at ϕ=0, as can clearly be seen in Figure 2d. For ϕ=π/2, we always have more TM photons emitted perpendicular to the plate, while for ϕ=0 this is true only for small frequencies, with the maximum shifting towards θ=π/2 in the high-frequency limit ($\omega \to \omega_{\mathrm{cm}}$).

We find the frequency spectrum by performing the angular integration of (Equation 35):(38)$$\begin{array}{c} \frac{d\Gamma_{\sigma }^{\|}}{d\omega }=\frac{8\omega^{3}{\left(\omega_{\mathrm{cm}}-\omega \right)}^{3}v^{2}R^{6}}{405\pi^{3}c^{8}}F_{\sigma }^{\|}\left(\frac{\omega }{\omega_{\mathrm{cm}}}\right)\;, \end{array}$$
with
(39)$$\begin{array}{ccc} F_{\mathrm{TE}}^{\|}\left(x\right) & = & 19x^{2}-23x+29 \end{array}$$
(40)$$\begin{array}{ccc} F_{\mathrm{TM}}^{\|}\left(x\right) & = & 5x^{2}-x+3\;. \end{array}$$

Differently to the previous case of motion perpendicular to the surface of the mirror, here there are more TE than TM photons for any portion of the spectra, as illustrated by Figure 4. Also, in contrast with the previous case, the TM spectrum favors emission at $\omega >\omega_{\mathrm{cm}}/2$, with the opposite happening for TE photons. The total spectrum is again symmetrical around $\omega =\omega_{\mathrm{cm}}/2$, as it should be.

The photon production rates for each polarization are derived by integrating the spectra over all frequencies:(41)$$\Gamma_{\mathrm{TE}}^{\|}=\frac{82\omega_{\mathrm{cm}}^{7}v^{2}R^{6}}{25,515\pi^{3}c^{8}}=\frac{41}{7}\Gamma_{\mathrm{TM}}^{\|}\;.$$
The total rate $\Gamma^{\|}=\Gamma_{\mathrm{TE}}^{\|}+\Gamma_{\mathrm{TM}}^{\|}$ is smaller than $\Gamma^{⊥}$ by a factor of 6/11, in agreement with the general discussion of (Equation 25).

### 4.3. Motion along Arbitrary Directions

The total frequency spectrum in the case of motions along a fixed but otherwise arbitrary direction can now be derived from the results obtained in the previous subsections for the two particular cases of perpendicular and parallel motions. First, we note that the dependence of the total spectrum on the direction of motion is entirely contained in the second term of the r.-h.-s. of Equation (Equation 24). Using the polarizability tensor of the disk (Equation 27), we find that this term is proportional to ${\left(\overset{↔}{\alpha }\hat{u}\right)}^{2}=\alpha_{\|}^{2}\sin^{2}\chi ,$ where $\alpha_{\|}=\alpha_{xx}=\alpha_{yy}$ and χ is the angle between the direction of motion and the disk’s symmetry axis. Since the above contribution vanishes for perpendicular motion (χ=0), the first term in the r.-h.-s. of Equation (Equation 24), proportional to ${\left(\overset{↔}{\alpha }\right)}^{2},$ is in fact the total spectrum $\frac{d\Gamma^{⊥}}{d\omega },$ while the total spectrum for parallel motion $\frac{d\Gamma^{\|}}{d\omega }$ is the sum of the two contributions appearing in Equation (Equation 24) with χ=π/2. Thus, the spectrum for motion along an arbitrary direction is given by
(42)$$\frac{d\Gamma }{d\omega }=\frac{d\Gamma^{⊥}}{d\omega }+\left(\frac{d\Gamma^{\|}}{d\omega }-\frac{d\Gamma^{⊥}}{d\omega }\right)\sin^{2}\chi \;.$$

Note that the total spectra $\frac{d\Gamma^{⊥}}{d\omega }$ and $\frac{d\Gamma^{\|}}{d\omega }$ can be obtained by adding the results for the two orthogonal polarizations given in Section 4.1 and Section 4.2, respectively. The total emission rate can also be written in terms of the total rates for perpendicular and parallel motion in the same form of Equation (Equation 42). On the other hand, no simple expression holds for the angular distribution nor for the frequency spectra discriminated by polarization. Similar comments also hold for our next example.

## 5. Thin Cylindrical Metallic Rod

As a final example we shall consider a thin cylindrical metallic rod of length *L* and small radius ρ≪L, as depicted in Figure 5. Denoting its symmetry axis by *z*, the dominant contribution to the polarizability comes from the element [40]$\alpha_{zz}=L^{3}/\left(24\log\left(2L/\rho \right)-56\right)$. Note that even this element vanishes in the limit ρ→0, indicating that the DCE in this situation is far smaller than in the case of a flat mirror discussed in Section 4, as expected, since we now have essentially a one-dimensional scatterer. As in the previous section, we first evaluate the angular distribution from the general result (Equation 22).

As usual when dealing with cylindrical symmetry, we define the transverse electric polarization with the condition that the electric field is orthogonal to the rod’s symmetry axis. We then define the unit vectors ${\hat{\epsilon }}_{k}^{\left(\mathrm{TE}\right)}=\hat{z}\times \hat{k}/\left|\hat{z}\times \hat{k}\right|$ and ${\hat{\epsilon }}_{k}^{\left(\mathrm{TM}\right)}={\hat{\epsilon }}_{k}^{\left(\mathrm{TE}\right)}\times \hat{k}$ as the unit vectors for TE and TM polarizations, respectively. The angular spectra are generally given by
(43)$$\frac{d\Gamma_{\sigma }^{⊥,\|}\left(k\right)}{d\omega d\Omega }=\frac{\omega^{3}{\left(\omega_{\mathrm{cm}}-\omega \right)}^{3}v^{2}\alpha_{zz}^{2}}{60\pi^{2}c^{8}}f_{\sigma }^{⊥,\|}\left(\frac{\omega }{\omega_{\mathrm{cm}}},\theta ,\phi \right)\;,$$
When the motion is perpendicular to the rod’s symmetry axis (see Figure 5a), the angular dependence is captured by the functions
(44)$$\begin{array}{ccc} f_{\mathrm{TE}}^{⊥}\left(x,\theta ,\phi \right) & = & 5\sin^{2}\phi \cos^{2}\theta \end{array}$$
(45)$$\begin{array}{ccc} f_{\mathrm{TM}}^{⊥}\left(x,\theta ,\phi \right) & = & 5\cos^{2}\phi {\left(1-x\sin^{2}\theta \right)}^{2}+\sin^{2}\theta \left(2x^{2}+x+2\right)\;. \end{array}$$
where θ is the angle between k and the rod’s symmetry axis (*z*-axis), while ϕ denotes the angle between the direction of motion (unit vector $\hat{u}=\hat{x}$) and the projection of k on the plane perpendicular to the rod (xy plane).

At θ=0, the spectra for TE and TM polarizations are proportional to $\sin^{2}\phi$ and $\cos^{2}\phi$, respectively. Analogous with the case of a disk moving sideways discussed in Section 4.2, such dependence on ϕ results from a vector decomposition on the polarization basis in the limit θ→0. The unit vector along the direction of motion reads $\hat{x}=-\sin\phi \;{\hat{\epsilon }}_{k}^{\left(\mathrm{TE}\right)}+\cos\phi \;{\hat{\epsilon }}_{k}^{\left(\mathrm{TM}\right)}.$ Thus, Equations (44) and (45) show that the DCE radiation emitted parallel to the symmetry axis (θ=0) is linearly polarized along the direction of motion, in contrast with the analog case for a disk, for which the radiation is partially polarized along the direction perpendicular to the motion.

As $f_{\mathrm{TE}}^{⊥}$ does not depend on *x*, the frequency dependence of the spectrum for TE polarization is entirely contained in the pre-factor appearing in (Equation 43), which arises from the electromagnetic density of states. Such a property can also be seen in Equation (Equation 21) since $\alpha_{ij}{\hat{\epsilon }}_{kTE}^{i}=0$. In other words, only the Röntgen current contributes to the emission of TE-polarized photons.

The plane ϕ=0, defined by $\hat{u}$ and by the rod’s axis, is a symmetry plane with respect to space reflection. Therefore, dynamical Casimir photons emitted at ϕ=0 are necessarily TM-polarized by symmetry (see Section 4.2 for a similar discussion concerning a flat mirror moving sideways).

Equation (44) also shows that TE-polarized photons are mostly emitted close to the direction of the rod’s axis, as illustrated by Figure 6a for the case $\omega =\omega_{\mathrm{cm}}/2$. In addition, for any given θ, the emission is maximal along the direction perpendicular to the motion (ϕ=π/2).

As for the TM angular spectrum, it has a maximum at ϕ=0 for any given value of θ. On the plane ϕ=0, the TM angular spectrum for frequencies $\omega >2\omega_{\mathrm{cm}}/7$ is maximal along the direction parallel to the rod’s axis, as for example in the case $\omega =\omega_{\mathrm{cm}}/2$ shown in Figure 6b, while lower frequency photons are mostly emitted parallel to the direction of motion (θ=π/2).

Finally, when the rod is moving parallel to its axis (see Figure 5b), the angular spectrum is axially symmetric and hence independent of ϕ:(46)$$\begin{array}{ccc} f_{\mathrm{TE}}^{\|}\left(x,\theta \right) & = & 0 \end{array}$$(47)$$\begin{array}{ccc} f_{\mathrm{TM}}^{\|}\left(x,\theta \right) & = & 5\sin^{2}\theta \cos^{2}\theta x^{2}+\sin^{2}\theta {\left(1-x\right)}^{2}\;, \end{array}$$
In this configuration, any plane containing the rod is a symmetry plane, thus explaining the complete absence of TE photons. Note that the angular spectrum vanishes at θ=0, as expected since the results for TM and TE polarizations must coincide in this limit. Low-frequency photons are mostly emitted perpendicularly to the motion (θ=π/2). As the frequency increases, the angle of maximum emission decreases and approaches θ=π/4 as $\omega \to \omega_{\mathrm{cm}}.$ Figure 6c shows the angular spectrum for $\omega =\omega_{\mathrm{cm}}/2.$

Note that the angular distributions obtained for both the disk and the rod, illustrated by Figure 2 and Figure 6, respectively, are combinations of the familiar patterns for dipole and quadrupole radiation. The derivation of the Hamiltonian presented in Section 2 explains the origin of this resemblance. Indeed, the motion of the scatterer can be represented in terms of time-dependent electric quadrupole and magnetic dipole contributions.

The frequency spectra for both perpendicular and parallel motions have the form
(48)$$\frac{d\Gamma_{\sigma }^{⊥,\|}}{d\omega }=\frac{\omega^{3}{\left(\omega_{\mathrm{cm}}-\omega \right)}^{3}v^{2}\alpha_{zz}^{2}}{90\pi c^{8}}F_{\sigma }^{⊥,\|}\left(\frac{\omega }{\omega_{\mathrm{cm}}}\right)\;,$$
with
(49)$$\begin{array}{ccc} F_{\mathrm{TE}}^{⊥} & = & 5\; \end{array}$$
(50)$$\begin{array}{ccc} F_{\mathrm{TM}}^{⊥}\left(x\right) & = & 16x^{2}-16x+23\;, \end{array}$$
(51)$$\begin{array}{ccc} F_{\mathrm{TE}}^{\|} & = & 0\; \end{array}$$
(52)$$\begin{array}{ccc} F_{\mathrm{TM}}^{\|}\left(x\right) & = & 4(2x^{2}-2x+1)\;. \end{array}$$
In contrast to the case of a flat mirror discussed in Section 4, we see that each polarization spectrum is separately symmetrical with with respect to $\omega =\omega_{\mathrm{cm}/2}$ since $F_{\mathrm{TE}}^{⊥}$ is frequency independent. This property holds for any direction of motion and results from the fact that only the Röntgen current contributes to TE emission.

The total photon production rates are given by
(53)$$\begin{array}{ccc} \Gamma_{\mathrm{TM}}^{⊥} & = & \frac{\omega_{\mathrm{cm}}^{7}v^{2}\alpha_{zz}^{2}}{648\pi c^{8}}=\frac{35}{9}\Gamma_{TE}^{⊥}\;, \end{array}$$
(54)$$\begin{array}{ccc} \Gamma_{\mathrm{TM}}^{\|} & = & \frac{\omega_{\mathrm{cm}}^{7}v^{2}\alpha_{zz}^{2}}{5670\pi c^{8}}\;,\Gamma_{\mathrm{TE}}^{\|}=0. \end{array}$$
By comparing (Equation 53) with (54), we conclude that a motion parallel to the rod’s axis leads to 11 times less photons than a motion perpendicular to the axis. Such reduction factor can also be derived directly from the general result (Equation 25).

## 6. Final Remarks

We have analyzed the dynamical Casimir radiation produced by anisotropic scatterers moving in the quantum vacuum. For the paradigmatic case of an oscillating mirror, the values for the photon wavelength are typically much larger than the transverse size of the mirror. Thus, modelling the mirror as a surface of infinite extension is not realistic.

Here, we have taken the opposite limit, corresponding to the electric dipole approximation, which provides a much more accurate description in most cases. We have analyzed the DCE for macroscopic neutral bodies without a permanent electric dipole, which emit no classical radiation, and for which the electromagnetic response is fully captured by their (generally anisotropic) polarizability tensor. Following these considerations, we have built up an effective Hamiltonian to capture the DCE radiation by macroscopic and anistropic scatterers. This effective Hamiltonian is obtained through a multipolar expansion involving electric quadrupole and magnetic dipole contributions that result from the motion of the scatterer. Such Hamiltonian formalism for anisotropic scatterers enabled us to extend into the macroscopic realm a previous approach of DCE restricted so far to microscopic systems [23].

Considering a harmonic motion specifically, we obtained a general result for the DCE emission rate in terms of the polarizability tensor of the moving body. We find that the photon emission is maximum when the motion occurs along the principal axis corresponding to the smallest eigenvalue of this polarizability tensor. This result confirms the common intuition that DCE is maximal when the body moves along the direction of its smallest spatial extension, e.g., the normal to the surface of a moving disk.

We illustrated this approach in two examples that correspond to distinct geometries, namely a circular disk and a cylindrical rod. We have obtained the separate DCE contributions corresponding to the motion either parallel or perpendicular to the symmetry axis of each system. In particular, we have analyzed the most typical DCE configuration corresponding to the motion of a circular disk along its normal and compared the predictions of our dipole-approximation approach with the usual infinite plate model. Our findings reveal that the latter overestimates the DCE photon production rate by many orders of magnitude for any realistic configuration. There are also qualitative and conceptual differences: a finite-size disk can emit cross-polarized photon pairs, whereas photon pairs are necessarily co-polarized for an infinite plate model due to its translation symmetry. We have also discussed the manifestations of the translation symmetry breaking in the DCE angular and frequency spectra.

Finally, our Hamiltonian approach holds as long as the dipole approximation is applicable, which is typically the case in configurations involving finite-size objects undergoing a non-relativistic motion. Thus, our approach can be applied not only to DCE with an isolated single scatterer, but also to analyze more general QED effects involving the non-relativistic motion of one or several bodies.

## Figures and Tables

**Figure 1 entropy-26-00251-f001:**
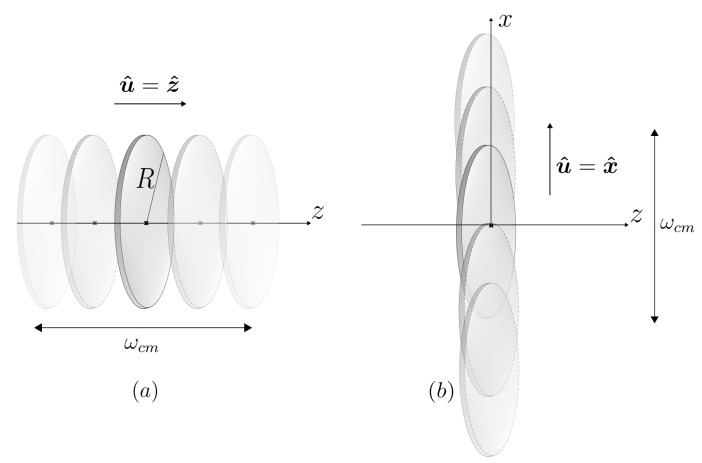
Metallic disk of radius *R* oscillating with frequency $\omega_{\mathrm{cm}}$ along a direction $\hat{u}$ either (**a**) perpendicular or (**b**) parallel to its surface.

**Figure 2 entropy-26-00251-f002:**
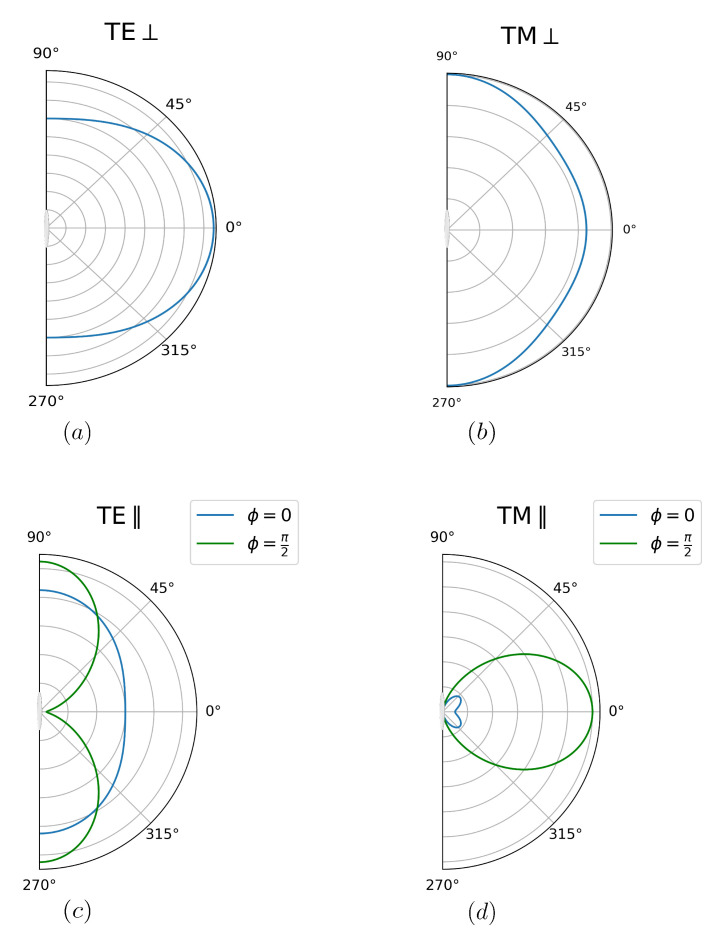
Polar plots representing the angular distribution for photon emission as a function of the emission angle θ, as measured from the normal to the plane containing the disk. All plots correspond to the photon frequency $\omega =\omega_{\mathrm{cm}}/2$. Panels (**a**) for TE and (**b**) for TM polarization refer to the case where the disk moves perpendicularly to the plane. This case has axial symmetry. The configuration with motion parallel to the plane is represented by (**c**) for TE and (**d**) for TM polarization. The angular distributions now depend also on the azimuthal angle ϕ. The plots in (**c**,**d**) correspond to ϕ=0 (blue) and ϕ=π/2 (green). The direction of motion corresponds to θ=π/2 and ϕ=0.

**Figure 3 entropy-26-00251-f003:**
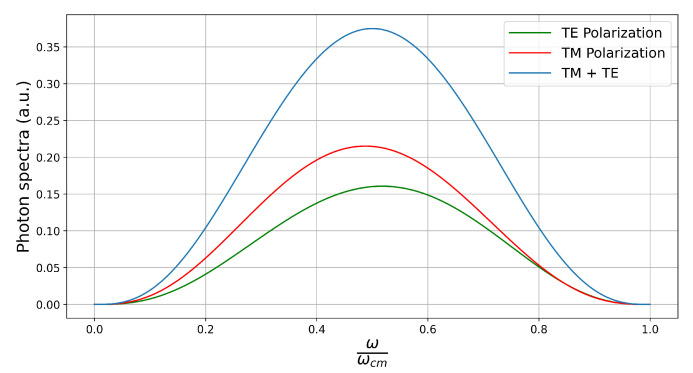
Frequency spectra for TE- (green) and TM-polarized (red) photons as functions of $\omega /\omega_{\mathrm{cm}}$. The blue line represents the sum of the TE and TM spectra and is symmetrical around $\omega_{\mathrm{cm}}/2$. The disk is moving perpendicularly to its plane.

**Figure 4 entropy-26-00251-f004:**
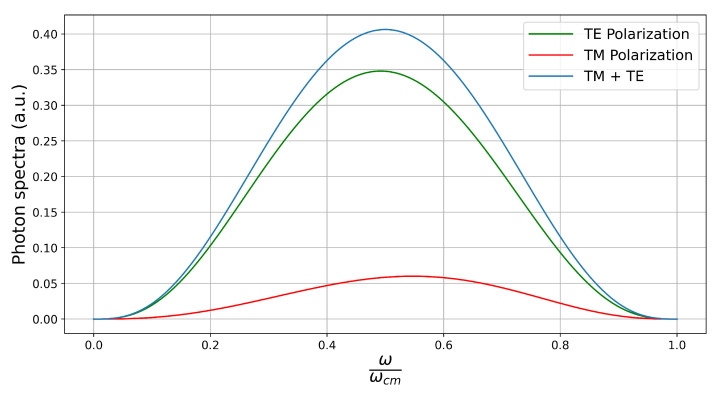
Same conventions as in Figure 3. The disk is moving parallel to the plane.

**Figure 5 entropy-26-00251-f005:**
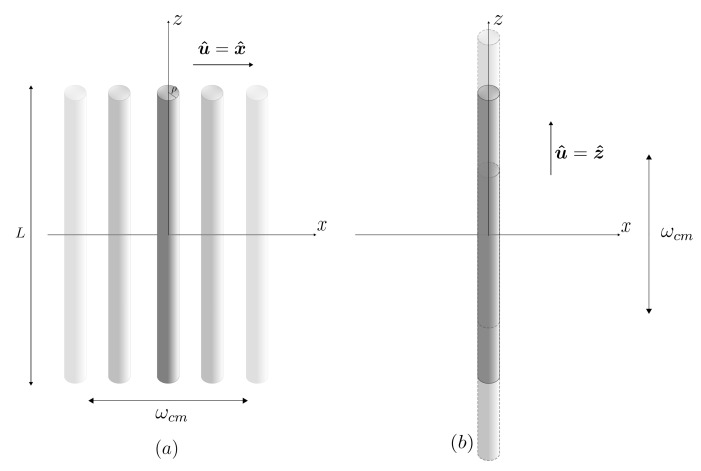
Thin cylindrical rod of length *L* oscillating with frequency $\omega_{\mathrm{cm}}$ along a direction $\hat{u}$ either (**a**) perpendicular or (**b**) parallel to its axis.

**Figure 6 entropy-26-00251-f006:**
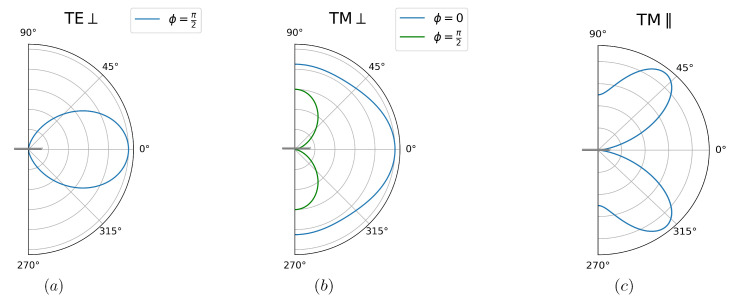
Polar plots representing the angular distribution for photon emission as function of the emission angle θ, as measured from the rod’s axis. All plots correspond to the photon frequency $\omega =\omega_{\mathrm{cm}}/2$. Panels (**a**) for TE and (**b**) for TM polarization refer to motion perpendicular to the symmetry axis. The angular distributions depend also on the azimuthal angle ϕ. The TE distribution vanishes at ϕ=0, and we take ϕ=π/2 in panel (**a**). The plots in (**b**) correspond to ϕ=0 (blue) and ϕ=π/2 (green). The direction of motion corresponds to θ=π/2 and ϕ=0. (**c**) Axially symmetric case of motion parallel to the axis, for which there is no TE emission.

## Data Availability

No new data were created or analyzed in this study.

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
