# Peer review of "Multipole Approach to the Dynamical Casimir Effect with Finite-Size Scatterers"

_entropy, 2024, doi:10.3390/e26030251_

Round 1
Reviewer 1 Report
Comments and Suggestions for Authors
Please read the attached file.

Author Response
See attached pdf file.

Reviewer 2 Report
Comments and Suggestions for Authors
The authors studied the dynamical Casimir effect with anisotropic scatterers. The new approach—Hamiltonian approach—is exploited to calculate the DCE by a moving anisotropic scatterer within the dipolar approximation. This approach has no need to assume the infinite metallic surface, which is necessary condition for the traditional ap- proach to learn the DCE. This extension could give a more realistic results compared with the traditional one. As pointed out by the authors, the infinite plate calculation over- estimates the photon production rate by many orders. Therefore, this approach is very important for learning the DCE. The authors considered two examples as an application of their formalism: a finite-size mirror oscillating at a typical mechanical frequency; the dynamical Casimir photons produced by a metallic oscillating rod. Interesting results has been shown. The whole manuscript is well-written, and all the expressions are clear. Therefore, I recommend this manuscript for publication in Entropy journal. There is a comment:
1) it is more interesting to discuss the case where the disk oscillates along the direction neither perpendicular nor parallel to its surface, while with a fixed angle?
Round 2
Reviewer 1 Report
Comments and Suggestions for Authors
The authors have well replied all my previous comments. I recommend it for publication.